# UNITRACK: DIFFERENTIABLE GRAPH REPRESENTATION LEARNING FOR MULTI-OBJECT TRACKING

**Bishoy Galoaa**[*]**, Xiangyu Bai**[*]**, Utsav Nandi,**
**Sai Siddhartha Vivek Dhir Rangoju**, **Somaieh Amraee**, **Sarah Ostadabbas**
Augmented Cognition Lab (ACLab),
Electrical and Computer Engineering Department, Northeastern University
[*]Equal contribution

## ABSTRACT

We present UniTrack, a plug-and-play graph-theoretic loss function designed to significantly enhance multi-object tracking (MOT) performance by directly optimizing tracking-specific objectives through unified differentiable learning. Unlike prior graph-based MOT methods that redesign tracking architectures, UniTrack provides a universal training objective that integrates detection accuracy, identity preservation, and spatiotemporal consistency into a single end-to-end trainable loss function, enabling seamless integration with existing MOT systems without architectural modifications. Through differentiable graph representation learning, UniTrack enables networks to learn holistic representations of motion continuity and identity relationships across frames. We validate UniTrack across diverse tracking models and multiple challenging benchmarks, demonstrating consistent improvements across all tested architectures and datasets including Trackformer, MOTR, FairMOT, ByteTrack, GTR, and MOTE. Extensive evaluations show up to 53% reduction in identity switches and 12% IDF1 improvements across challenging benchmarks, with GTR achieving peak performance gains of 9.7% MOTA on SportsMOT. Code and additional resources are available at `https://github.com/ostadabbas/UniTrack` and `https://ostadabbas.github.io/unitrack.github.io/`.

## 1 INTRODUCTION

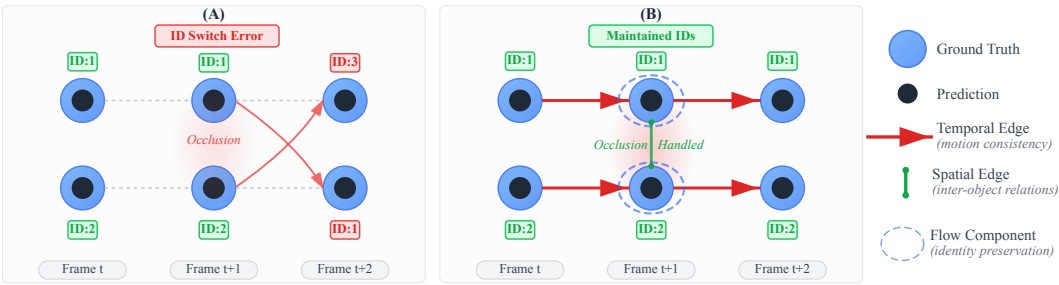

Figure 1: Comparison of UniTrack's graph-based approach and classical multi-object tracking. (A) A detection-based tracking handles trajectories independently, resulting in ID switches at occlusion: person 1 reassigned to ID 3 and person 2 to ID 1 (crossing arrows and red-highlighted boxes). (B) Our graph-based approach maintains correct identities through the same occlusion via three integrated components: temporal edges (red arrows) for motion consistency, spatial edges (green lines) for inter-object relationships, and flow components (blue dashed ellipses) for identity preservation. Green-highlighted ID boxes show successful identity maintenance throughout the sequence. Ground truth (blue circles) and predictions (dark centers) demonstrate how unified optimization prevents ID switches in challenging scenarios.

Multi-object tracking (MOT) is as a critical element of video-based learning, aiding extraction of detailed information about movement, interaction patterns, and spatial relationships in downstream

applications such as activity recognition, behavior analysis, and anomaly detection. A central objective of MOT is to accurately identify and consistently follow individuals across video frames, assigning unique identifiers to subjects throughout the sequence Zeng et al. (2021); Meinhardt et al. (2021); Zhang et al. (2021a); Wojke et al. (2017); Galoaa et al. (2025a). While the last decade has seen significant advancements in MOT methodologies, effectively tracking multiple interacting and occluded objects in diverse and complex environments continues to pose substantial challenges. Despite the advances from recent transformer-based approaches such as MOTR Zeng et al. (2021), TrackFormer Meinhardt et al. (2021), and GTR Zhou et al. (2022) and recent work on multi-camera tracking Galoaa et al. (2025c), inefficiencies stemming from the separation of detection and tracking in traditional approaches persist–particularly under challenging conditions like occlusions, crowded scenes, and complex motion dynamics.

Existing MOT techniques Bewley et al. (2016b); Wojke et al. (2017); Zhang et al. (2021a) utilize a mix of detection metrics such as IoU (intersection over union) Rezatofighi et al. (2019) and classification metrics like cross-entropy Meinhardt et al. (2021); Zeng et al. (2021) as their training objectives. These metrics do not adequately evaluate the complex interplay between temporal stability, spatial awareness, and identity preservation–elements vital for effective tracking Zhang et al. (2021b); Sun et al. (2020). Consequently, high detection accuracy models falter in maintaining consistent object identities, particularly through occlusion episodes Zhang et al. (2021a); Zeng et al. (2021). These limitations become especially pronounced in complex scenarios involving dense crowds, rapid motion, variable object scales, and congested environments Dendorfer et al. (2021), demanding robust mechanisms for preserving object identities. Our research reveals that there are three primary categories of errors that existing methods fail to address holistically: *post-occlusion ID switches*, where identity tracking fails when direct line-of-sight is obstructed (error Type 1), *temporal inconsistency*, where the tracker struggles to maintain stable ID assignments when subjects change posture (error Type 2), and *cross-subject ID switches*, where IDs are incorrectly exchanged when subjects cross paths and later separate, leading to tracking instability (error Type 3).

To overcome the noted limitations in MOT, we introduce UniTrack, a novel differentiable graph representation learning framework with an embedded loss function. This innovative approach leverages a hierarchical graph structure to model the spatial and temporal dynamics among tracked objects, integrating detection accuracy, spatial consistency, and temporal continuity into a singular, comprehensive learning objective. By enabling direct optimization of tracking-specific metrics through differentiable graph computations, UniTrack offers a robust and adaptable framework that seamlessly integrates with existing end-to-end MOT systems. Compared to to traditional independent trajectory approaches, UniTrack addresses three key error types through a joint optimization enabled by unified graph structure: flow components reduce Type 1 errors by maintaining identity consistency, temporal edges prevent Type 2 errors by enforcing motion consistency, and spatial edges mitigate Type 3 errors by modeling inter-object relationships, significantly reducing identity switches and enhancing overall stability (see Figure 1).

The versatility of UniTrack lies in its capability to dynamically re-weight tracking components, enabling automatic model tuning for particular environmental conditions without architectural changes. In crowded scenes, spatial consistency is prioritized to handle occlusions, while fast-motion settings will benefit from stronger temporal coherence. Our evaluations underscore the efficacy of UniTrack, showcasing enhancements over traditional methods on well-established benchmarks like MOT17 Milan et al. (2017), MOT20 Dendorfer et al. (2021), SportsMOT Cui et al. (2023) and DanceTrack Sun et al. (2022). Notably, our models excel in maintaining object identities and tracking precision, especially in occluded and crowded environments, highlighting the significant benefits of our unified approach. The key contributions of our are as follows:

- We introduce UniTrack, a novel graph-based representation learning framework unifying detection accuracy, identity preservation, and spatial-temporal consistency to directly address three key tracking error types.

- We develop an adaptive weighting mechanism using graph Laplacian analysis that automatically balances components based on scene characteristics without manual tuning.

- We integrate UniTrack with existing approaches (MOTR Zeng et al. (2021), TrackFormer Meinhardt et al. (2021), FairMOT Zhang et al. (2021b), ByteTrack Zhang et al. (2022), GTR Zhou et al. (2022), MOTE Galoaa et al. (2025b)) without architectural modifications.

- We conduct extensive experiments on MOT17 Milan et al. (2017), MOT20 Dendorfer et al. (2021), SportsMOT Cui et al. (2023), and DanceTrack Sun et al. (2022), demonstrate consistent reductions in ID switches, temporal inconsistency, and tracking failures, with GTR achieving up to 9.7% MOTA and 12.3% IDF1 improvements on the SportsMOT dataset.

## 2 RELATED WORK

**Classical Approaches in MOT.**
The training of MOT systems has traditionally

relied on a combination of separate tracking and detection modules. Most tracking frameworks optimize detection using standard object detection models such as YOLO Jiang et al. (2022) with IoU or generalized IoU (GIoU) as loss metrics, while handling tracking association through separate optimization objectives Bewley et al. (2016a); Wojke et al. (2017). This disjoint formulation often fails to capture the intricate relationships between detection and tracking performance, particularly during training phase, where end-to-end optimization is most beneficial. As shown in Figure 2, traditional approaches primarily focus on detection accuracy (partially addressing aspects of Type 1 errors) but struggle with preserving identities through occlusions Bergmann et al. (2019); Luo et al. (2021). Furthermore, they lack explicit mechanisms for enforcing temporal consistency (Type 2 errors) and modeling inter-object relationships (Type 3 errors), leading to unstable ID assignments when subjects interact or change appearance Ristani & Tomasi (2018); Milan et al. (2014). Recent work has introduced more sophisticated approaches to enhance MOT training and address these challenges: ByteTrack Zhang et al. (2021a) utilizes detection confidence as a part of the loss function, though primarily focusing on inference-time association rather than training optimization. TransTrack Sun et al. (2020) introduces memory-based temporal modeling, yet lacks a unified framework that could optimize all tracking components simultaneously. **Joint Detection-Tracking Approaches.** Recent algorithms explored integrated training for end-to-end MOT systems. MOTR Zeng et al. (2021) incorporates a track query matching function

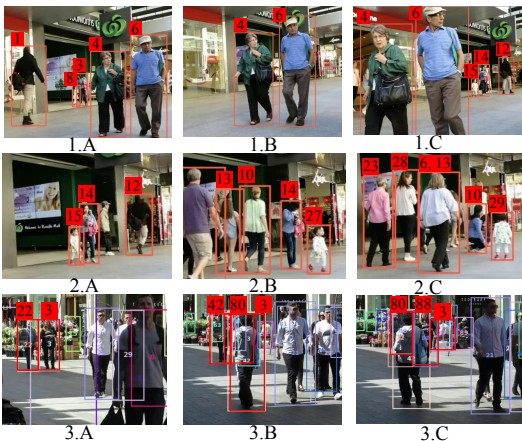

Figure 2: Illustration of tracking errors in MOTR Zeng et al. (2021) on MOT17 sequences 8 and 9. The first row (sequence 9) highlights post-occlusion ID switches (error Type 1): subject 1 loses tracking when occluded behind subject 4 in frame 1.B, with IDs 8, 3, and 1 subsequently reassigned as IDs 15, 14, and 12 in frame 1.C. The second row (sequence 8) demonstrates temporal inconsistency (error Type 2), where the tracker fails to maintain IDs when subjects change postures: ID 15 changes to 27 (2.B), and in 2.C, MOTR erroneously assigns two bounding boxes to subject 6, 13 while ID 14 changes to 10 and ID 15 is reassigned to 29, illustrating instability in temporal association. The third row (sequence 9) demonstrates cross-subject ID switches (error Type 3): ID 22 and 3 are correctly assigned in 3.A, but when subject 42 occludes subject 3 in 3.B, it triggers a cascade of errors–ID 22 gets incorrectly swapped with ID 80, followed by ID 3 being erroneously reassigned as ID 88 in 3.C, showcasing how occlusions propagate tracking failures.

alongside detection supervision in its training objective. Similarly, TrackFormer Meinhardt et al. (2021) proposes losses for track initialization and propagation during training. However, these tracking methods still largely treat spatial and temporal consistency as separate components, limiting the potential for truly integrated joint optimization during the training phase. While these approaches advance the handling of Type 1 errors such as post-occlusion identity switches via improved detection-to-tracking associations, their reliance on separate formulations continues to fall short in addressing Type 2 (temporal inconsistency) and Type 3 (cross-subject identity switches) errors, especially in complex scenarios.

**Graph-based MOT Methods.** Prior works have explored graph representations in MOT algorithms. Neural MOT solver Brasó & Leal-Taixé (2020) use message passing networks for data association,

treating tracking as edge classification. Recent approaches including SUSHI Cetintas et al. (2023), Global Tracking Transformers (GTR) Zhou et al. (2022), and DiffMOT Luo et al. (2024) incorporate graph structures into their tracking pipelines. Crucially, our proposed UniTrack framework differs fundamentally from these architectural approaches. While prior graph-based methods redesign the tracking algorithm itself—requiring modifications to network architecture, forward pass logic, and inference procedures—UniTrack provides a graph-theoretic *loss function* that serves as a universal training objective. This distinction is central to understanding our contribution: UniTrack is not an architectural innovation but rather a plug-and-play training enhancement that can be integrated into any existing MOT system by simply adding our loss term during training, with zero modifications to model architecture or inference code. As demonstrated in our experiments across 7 diverse architectures (transformer-based, tracking-by-detection, joint detection-tracking, and memory-augmented), this universal applicability distinguishes UniTrack as a training-time enhancement rather than a specialized architectural design.

## 3 UniTrack: A Graph-Theoretic Learning Framework

**Problem Formulation**–Given a video sequence of $T$ frames, we aim to learn a tracking model that optimizes both detection accuracy and tracking consistency through unified differentiable representation and objective. Our approach specifically addresses three key error types: post-occlusion ID switches (Type 1), temporal inconsistency (Type 2), and cross-subject ID switches (Type 3). Traditional tracking approaches handle trajectories independently, leading to identity switches when objects intersect, while UniTrack leverages unified graph structures with spatial-temporal connections for consistent tracking. We formulate this as a graph-based optimization problem where the loss function $\mathcal{L}$ is defined over a temporal sequence of weighted directed graphs:

$$\mathcal{G} = \{G_t = (V_t, E_t, W_t)\}_{t=1}^T, \tag{1}$$

where $V_t = \{v_t^i | i \in \mathcal{I}_t\}$ is the set of all vertices at time $t$, with $v_t^i$ representing tracked object $i$ at time $t$. The set $\mathcal{I}_t$ contains all object identities present at time $t$. Edges $E_t$ capture spatial-temporal relationships between objects across consecutive frames, and weights $W_t = \{w_t^{ij} | (i, j) \in E_t\}$ encode confidence and association strengths for each potential object correspondence. The graph construction enables joint optimization across all objects and time steps, forming a structured flow optimization framework that maintains both local coherence and global consistency.

### 3.1 Graph Flow Network Architecture

We model the tracking problem as a flow network, where the graph structure encodes the complex relationships inherent in multi-object tracking. The key insight is that object tracking can be viewed as a flow conservation problem: objects cannot arbitrarily appear or disappear, and each detection should correspond to at most one physical object across time. To capture the life-cycle of tracked objects, we introduce balance variables $b_t^i \in \{-1, 0, 1\}$ for each object $i$ at time $t$, where $b_t^i = 1$ indicates track initialization (new object appearing), $b_t^i = 0$ indicates track continuation (existing object persisting), and $b_t^i = -1$ indicates track termination (object disappearing from view). These balance variables encode the net flow change for each object at each time step, ensuring that the flow conservation constraints maintain physically consistent tracking states throughout the sequence.

We introduce flow variables $f_t^{ij}$ representing the association strength between object $i$ at time $t$ and object $j$ at time $t + 1$. To maintain physical consistency in tracking, we enforce flow conservation constraints at each node:

$$\sum_{j \in \mathcal{N}^+(i)} f_t^{ij} - \sum_{k \in \mathcal{N}^-(i)} f_{t-1}^{ki} = b_t^i, \quad \forall i \in V_t, \tag{2}$$

where $\mathcal{N}^+(i)$ is the set of nodes that can receive flow from node $i$ (outgoing neighbors), and $\mathcal{N}^-(i)$ is the set of nodes that send flow to node $i$ (incoming neighbors). Here, $j$ indexes over all possible destination objects for flow leaving object $i$, while $k$ indexes over all possible source objects sending flow to object $i$. This constraint ensures that the total outgoing flow minus incoming flow for each object equals its balance variable, naturally encoding object appearance, persistence, and disappearance. We construct the graph using a sliding window of $W = 5$ frames to balance computational efficiency with temporal context. The differentiable graph loss assembly follows three steps: (1)

construct node embeddings from detection features, (2) compute pairwise similarities to form edge weights and flow variables, and (3) apply flow conservation constraints while optimizing the unified loss. This process integrates seamlessly with backpropagation, enabling end-to-end training.

## 3.2 Unified Loss Components

Our loss function unifies three complementary components in a differentiable framework:

$$\mathcal{L} = \mathcal{L}_{\text{flow}} + \lambda_s \mathcal{L}_{\text{spatial}} + \lambda_t \mathcal{L}_{\text{temporal}}, \tag{3}$$

where $\lambda_s$ and $\lambda_t$ are adaptive weights automatically determined by scene characteristics (detailed in Section 3.3).

The flow-based loss $\mathcal{L}_{\text{flow}}$ primarily addresses Type 1 errors by encouraging confident object associations while adapting to detection quality:

$$\mathcal{L}_{\text{flow}} = - \sum_{(i,j) \in E_t} w^{ij} f_t^{ij} \cdot \exp\left(-\alpha \frac{|\mathcal{FP}|}{|\mathcal{P}|} - \alpha \frac{|\mathcal{FN}|}{|\mathcal{GT}|}\right), \tag{4}$$

where $\mathcal{FP}$ and $\mathcal{FN}$ represent false positive and false negative detections, $|\mathcal{P}|$ and $|\mathcal{GT}|$ denote total counts of predictions and ground truth objects, and $\alpha$ controls the influence of detection errors on the loss.

**Differentiability of Detection Quality Terms:** The FP and FN counts are computed from current predictions and ground truth at each training step. During backpropagation, these counts are treated as constants (stop-gradient), serving as adaptive coefficients that scale the loss magnitude based on detection performance. This approach maintains full differentiability with respect to flow variables $f_t^{ij}$-which directly receive gradients-while avoiding complications from differentiating discrete counting operations. The gradient with respect to flow variables is well-defined: $\frac{\partial \mathcal{L}_{\text{flow}}}{\partial f_t^{ij}} = -w^{ij} \exp\left(-\alpha \frac{|\mathcal{FP}|}{|\mathcal{P}|} - \alpha \frac{|\mathcal{FN}|}{|\mathcal{GT}|}\right)$, since $f_t^{ij} \in [0, 1]$ and the exponential function is smooth. When detection quality is high (low FP/FN rates), the exponential term approaches 1, fully trusting learned associations; when quality degrades, the term decreases, reducing commitment to uncertain associations.

The spatial coherence loss $\mathcal{L}_{\text{spatial}}$ targets Type 3 errors by enforcing that objects with similar spatial relationships should maintain consistent associations. This prevents identity switches when objects cross paths by considering the spatial context of nearby objects:

$$\mathcal{L}_{\text{spatial}} = \sum_{(i,j) \in E_t} w^{ij} d(\mathbf{p}_t^i, \mathbf{p}_{t+1}^j) f_t^{ij}, \tag{5}$$

where $\mathbf{p}_t^i = (x_t^i, y_t^i)$ represents the spatial coordinates of object $i$ at time $t$, $d(\cdot, \cdot)$ computes the geometric distance between positions across consecutive frames, and $w_{ij}$ are learned spatial attention weights. This term penalizes spatially distant associations to promote local coherence.

The temporal coherence loss $\mathcal{L}_{\text{temporal}}$ addresses Type 2 errors by ensuring smooth motion patterns and penalizing abrupt velocity changes that indicate tracking inconsistencies:

$$\mathcal{L}_{\text{temporal}} = \frac{1}{\Delta t} \sum_{i \in V_t} \|\mathbf{v}_t^i - \mathbf{v}_{t-1}^i\|_2^2 \sum_{j \in \mathcal{N}^+(i)} f_t^{ij}, \tag{6}$$

where $\mathbf{v}_t^i$ denotes the velocity vector of object $i$ at time $t$, computed from inter-frame position differences. The term $\Delta t$ represents the interval between consecutive frames and normalizes the temporal loss with respect to frame rate. This formulation penalizes sudden acceleration changes weighted by the confidence of the object's continued existence (sum of outgoing flows).

To account for varying problem scales, we apply logarithmic normalization to the final loss:

$$\mathcal{L}_{\text{final}} = \mathcal{L} \cdot \log(|\mathcal{E}| + 1), \tag{7}$$

where $|\mathcal{E}|$ represents the total number of valid edges, ensuring that the loss magnitude scales appropriately with scene complexity.

### 3.3 Adaptive Weight Learning

The relative importance of spatial and temporal components varies across different tracking scenarios. In crowded scenes, spatial consistency becomes crucial for handling occlusions, while fast-moving objects require greater emphasis on temporal coherence. To address this, we introduce an adaptive weighting mechanism that automatically adjusts $\lambda_s$ and $\lambda_t$ based on scene characteristics.

The key insight is that when object interactions are complex (indicating high spatial coupling), we should emphasize spatial consistency, while when motion patterns are erratic (indicating temporal challenges), we should emphasize temporal smoothness. We measure this complexity through the connectivity structure of our tracking graph using Laplacian analysis.

The adaptive weights are computed through graph Laplacian eigenvalue analysis, where the second smallest eigenvalue (algebraic connectivity) $\sigma_2(\mathbf{L})$ measures the connectivity of the components:

$$\lambda_s = \frac{\sigma_2(\mathbf{L}_s)^{-1}}{\sigma_2(\mathbf{L}_s)^{-1} + \sigma_2(\mathbf{L}_t)^{-1}}, \quad \lambda_t = \frac{\sigma_2(\mathbf{L}_t)^{-1}}{\sigma_2(\mathbf{L}_s)^{-1} + \sigma_2(\mathbf{L}_t)^{-1}}, \tag{8}$$

where $\mathbf{L}_s$ and $\mathbf{L}_t$ are the Laplacian matrices constructed from spatial and temporal edges respectively. Lower connectivity (smaller $\sigma_2$) indicates more fragmented relationships, requiring higher weighting to improve coherence. To provide mathematical insight into how the weights influence the system, we examine the partial derivatives of the total loss with respect to the weights (noting that the loss components $\mathcal{L}_{\text{spatial}}$ and $\mathcal{L}_{\text{temporal}}$ are already defined in Equations 5 and 6):

$$\frac{\partial \mathcal{L}}{\partial \lambda_s} = \mathcal{L}_{\text{spatial}}, \qquad \frac{\partial \mathcal{L}}{\partial \lambda_t} = \mathcal{L}_{\text{temporal}}. \tag{9}$$

However, $\lambda_s$ and $\lambda_t$ are not learnable parameters and cannot be updated via gradient descent (i.e., not: $\lambda_s^{(k+1)} = \lambda_s^{(k)} - \eta \frac{\partial \mathcal{L}}{\partial \lambda_s}$). Instead, they are recomputed at each training step from the current graph structure using Equation 8. This ensures the weights always reflect scene-specific graph connectivity rather than accumulating gradient-based updates. The model parameters $\theta$ are updated via standard backpropagation:

$$\theta^{(k+1)} = \theta^{(k)} - \eta \frac{\partial \mathcal{L}}{\partial \theta}, \tag{10}$$

where the gradients flow through the loss weighted by the current $\lambda_s$ and $\lambda_t$ values. As $\theta$ evolves, the resulting embeddings change the graph structure, which in turn updates the Laplacian matrices and recomputes new weight values. This mechanism enables automatic adaptation: when spatial relationships are fragmented (low $\sigma_2(\mathbf{L}_s)$), $\lambda_s$ increases to emphasize spatial consistency; when temporal flows are disrupted (low $\sigma_2(\mathbf{L}_t)$), $\lambda_t$ increases to enforce smoother motion patterns. We validate this recomputation approach against fixed and learned weight alternatives in Section 4.5 (Table 4).

### 3.4 Theoretical Analysis of Convergence

Our unified loss formulation provides theoretical guarantees for both optimization convergence and tracking consistency. We establish these properties to ensure UniTrack can be reliably integrated into existing MOT systems.

**Theorem 1 (Unified Convergence and Consistency)** *The UniTrack loss function $\mathcal{L} = \mathcal{L}_{\text{flow}} + \lambda_s \mathcal{L}_{\text{spatial}} + \lambda_t \mathcal{L}_{\text{temporal}}$ satisfies differentiability, local convergence under standard regularity conditions, and ensures physically plausible tracking solutions via flow conservation constraints.*

*Proof and Analysis:* The full theoretical analysis, including convergence conditions, differentiability proofs, and tracking consistency properties, is provided in Appendix A.1. These theoretical properties ensure that UniTrack not only improves tracking performance empirically but also provides a principled framework that maintains mathematical consistency throughout the optimization process.

## 4 Experimental Results

We conduct extensive experiments to evaluate the effectiveness of the loss function provided in UniTrack for existing MOT frameworks. Our approach is designed to enhance identity preservation

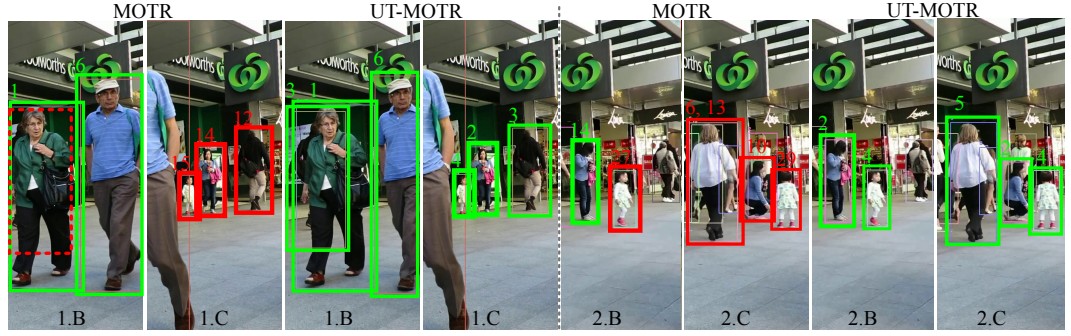

Figure 3: Comparative performance of MOTR Zeng et al. (2021) and UT-MOTR revisiting the same challenging scenarios from Figure 2. Red bounding boxes indicate tracking errors, green boxes show successful tracking, and dotted red boxes highlight missed detections. UT-MOTR successfully addresses the three error types demonstrated in Figure 2: maintaining consistent IDs through occlusions (frames 1.B-1.C), preserving temporal consistency during posture changes (frames 2.B-2.C), and preventing cross-subject ID switches in crowded scenes. The unified graph-theoretic approach enables robust tracking where the baseline MOTR fails. Additional qualitative results with Trackformer are provided in Section A.3.

while maintaining the core strengths of each baseline architecture, as demonstrated through comprehensive evaluations on MOT17 Milan et al. (2017), MOT20 Dendorfer et al. (2021), SportsMOT Cui et al. (2023), and DanceTrack Sun et al. (2022) benchmarks (as seen in Figure 3). Additionally, we analyzed the impact of each component on UniTrack's performance (see Figure 4 and Table 4).

## 4.1 IMPLEMENTATION DETAILS

UniTrack is implemented as an additional loss component that addresses three key MOT error types through its unified graph structure. For all baselines, we maintain their original hyperparameters and training protocols, with UniTrack's components weighted through our adaptive mechanism. The weighting parameter $\alpha$ in Equation 4 was set to 0.9, and we use a temporal window of 5 frames for graph construction. The adaptive weight learning rates start at $\eta = 0.01$ and decay following the baseline models' schedules. Comprehensive sensitivity analysis of all hyperparameters ($\tau$, $W$, $\Delta t$) with computational cost evaluation is provided in Appendix A.4. Memory requirements increase by approximately 5% due to graph construction, with computational complexity of $O(n^2 t)$ for $n$ objects over $t$ frames only for training. We follow standard evaluation protocols using HOTA Luiten et al. (2021), IDF1 Ristani et al. (2016), MOTA Keni Bernardin (2008), and identity switches (IDS) metrics.

Table 1: Performance comparison on challenging sequences that highlight the three error types. UT variants generally outperform their respective baselines across most metrics, with improvements varying by tracking architecture.

| Sequence | Method | MOTA↑ | IDF1↑ | HOTA↑ | FN↓ | IDs↓ |
|---|---|---|---|---|---|---|
| MOT17-13 | Trackformer | 57.42 | 63.16 | 66.29 | 1742 | 224 |
|  | **UT**-Trackformer | **62.86** | **67.38** | **68.45** | **1425** | **186** |
|  | MOTR | 58.63 | 62.47 | 65.18 | 1698 | **164** |
|  | **UT**-MOTR | **61.75** | **65.26** | **67.21** | **1536** | 172 |
| *Post-occlusion* | FairMOT | 56.98 | 61.84 | 64.76 | 1822 | 208 |
|  | **UT**-FairMOT | **60.43** | **64.58** | **66.92** | **1678** | **185** |
| MOT17-10 | Trackformer | 65.86 | 62.84 | 59.72 | 2356 | 113 |
|  | **UT**-Trackformer | **68.24** | **65.93** | **63.18** | **2087** | **104** |
|  | MOTR | 64.59 | 63.12 | 60.04 | 2427 | **84** |
|  | **UT**-MOTR | **67.31** | **66.25** | **62.38** | **2156** | 102 |
| *Temporal inconsis.* | FairMOT | 65.02 | 62.57 | 58.93 | 2392 | 108 |
|  | **UT**-FairMOT | **67.95** | **65.21** | **61.64** | **2134** | **98** |
| MOT17-02 | Trackformer | 51.23 | 37.51 | 42.63 | 5895 | 156 |
|  | **UT**-Trackformer | **54.76** | **44.28** | **48.12** | **5103** | **143** |
|  | MOTR | 50.46 | 39.25 | 43.48 | 5967 | 143 |
|  | **UT**-MOTR | **53.87** | **45.36** | **47.95** | **5287** | **132** |
| *Cross-subject* | FairMOT | 49.85 | 38.76 | 42.15 | 6054 | 162 |
|  | **UT**-FairMOT | **52.94** | **43.89** | **46.83** | **5465** | **149** |

## 4.2 UNITRACK ADDED TO EXISTING MOT

We selected diverse baseline architectures to demonstrate UniTrack's architectural agnosticism: Trackformer Meinhardt et al. (2021), MOTR Zeng et al. (2021), FairMOT Zhang et al. (2021b), ByteTrack Zhang et al. (2022), MOTE Galoaa et al. (2025b), and GTR Zhou et al. (2022). This diversity spans transformer-based end-to-end tracking (Trackformer, MOTR), joint detection-tracking (FairMOT), tracking-by-detection with association strategies (ByteTrack), and detection-embedding paradigms (GTR, MOTE). This selection allows us to validate UniTrack's effectiveness across different tracking approaches and demonstrate its plug-and-play nature. Each architecture represents

Table 2: Performance evaluation on MOT17 Milan et al. (2017) and MOT20 Dendorfer et al. (2021) test sets. Results compare standard tracking methods with and without UniTrack (UT) loss, ordered by publication year. Models trained with UT show significant improvements in detection and identity preservation, especially in MOTA and IDF1 scores. UniTrack significantly reduces false positives (FP) and false negatives (FN) while maintaining tracking stability. Performance is consistent between MOT17 and MOT20 results, demonstrating UniTrack's robustness. Bold values highlight performance gains achieved by integrating UniTrack, while shaded values denote the top-performing tracker among all contestants. We use this system for all tables unless stated otherwise.

| | MOT17 | | | | | | MOT20 | | | | | |
|---|---|---|---|---|---|---|---|---|---|---|---|---|
| Method | MOTA↑ | IDF1↑ | HOTA↑ | FP↓ | FN↓ | IDs↓ | MOTA↑ | IDF1↑ | HOTA↑ | FP↓ | FN↓ | IDs↓ |
| FairMOT Zhang et al. (2021b) | 61.7 | 61.5 | 52.9 | 1902 | 20456 | **388** | 53.5 | 58.3 | 52.4 | 1902 | 25456 | 488 |
| **UT**-FairMOT | **64.5** | **64.2** | **55.3** | **1623** | **19234** | 482 | **55.2** | **61.5** | **55.8** | **1723** | **23234** | **402** |
| MOTR Zeng et al. (2021) | 62.1 | 61.3 | 53.2 | 1843 | 21034 | 289 | 53.2 | 57.9 | 51.8 | 1843 | 25034 | 389 |
| **UT**-MOTR | **64.8** | **63.9** | **55.7** | **1562** | **19845** | **356** | **55.8** | **60.4** | **54.2** | **1562** | **23845** | **356** |
| Trackformer Meinhardt et al. (2021) | 62.3 | 57.6 | 52.8 | 1965 | 21893 | **643** | 54.1 | 56.2 | 50.9 | 1965 | 25893 | 643 |
| **UT**-Trackformer | **65.9** | **66.4** | **56.2** | **1039** | **16667** | 705 | **56.2** | **64.1** | **57.7** | **1374** | **22004** | 314 |
| ByteTrack Zhang et al. (2022) | 80.3 | 77.3 | 63.1 | 25491 | 83721 | 2196 | 77.8 | 75.2 | 61.3 | 26249 | 87594 | 1223 |
| **UT**-ByteTrack | **82.1** | **79.8** | **65.4** | **22145** | **79350** | **1865** | **79.5** | **77.8** | **63.7** | **23520** | **83145** | **1045** |
| GTR Zhou et al. (2022) | 75.3 | 71.5 | 59.1 | 1250 | 15800 | 1445 | 63.6 | 52.3 | 42.6 | 9916 | 205166 | 8604 |
| **UT**-GTR | **79.1** | **74.8** | **67.9** | **980** | **14200** | **951** | **63.8** | **52.5** | **43.0** | **9885** | **204530** | **8570** |
| MOTE Galoaa et al. (2025b) | 82.0 | 80.3 | 66.3 | 1100 | 8500 | 620 | 81.7 | 79.8 | 65.8 | 7800 | 12000 | 685 |
| **UT**-MOTE | **84.5** | **83.5** | **68.2** | **850** | **7200** | **542** | **83.2** | **81.4** | **67.1** | **7200** | **10500** | **578** |

a distinct philosophy in MOT: MOTR and Trackformer leverage transformer architectures for end-to-end tracking, FairMOT balances detection and re-identification in a single network, while GTR employs global tracking transformers and MOTE introduces enhanced multi-object tracking capabilities. By showing consistent improvements across these varied approaches, we establish UniTrack's generalizability as a universal enhancement for MOT systems.

### 4.3 Performance on Existing Benchmarks

The integration of UniTrack shows consistent and substantial improvements across different architectural designs when tested on multiple challenging datasets. Table 2 presents results on MOT17 and MOT20, while Table 3 shows performance on SportsMOT and DanceTrack.

**MOT17 and MOT20.** Most notably, when combined with any of these SOTA algorithms, Uni-

Table 3: Performance evaluation on SportsMOT Cui et al. (2023) and DanceTrack Sun et al. (2022) test sets. UniTrack demonstrates significant improvements on both datasets, with notable gains in identity preservation (IDF1) and ID switch reduction. SportsMOT presents challenges with rapid movements and complex interactions in sports scenarios, while DanceTrack features frequent occlusions, similar appearances, and complex dance movements. The consistent improvements across both datasets highlight UniTrack's effectiveness in handling diverse tracking scenarios. Bold value indicates performance gains from UniTrack integration; shading marks the best overall tracker.

| | SportsMOT | | | | | | DanceTrack | | | | | |
|---|---|---|---|---|---|---|---|---|---|---|---|---|
| Method | MOTA↑ | IDF1↑ | HOTA↑ | FP↓ | FN↓ | IDs↓ | MOTA↑ | IDF1↑ | HOTA↑ | FP↓ | FN↓ | IDs↓ |
| FairMOT Zhang et al. (2021b) | 90.8 | 53.5 | 49.3 | 8765 | 7234 | 2845 | 82.2 | 40.8 | 39.7 | 11234 | 16890 | 2987 |
| **UT**-FairMOT | **92.5** | **56.2** | **52.1** | **7890** | **6456** | **2234** | **84.8** | **43.5** | **42.3** | **10456** | **15234** | **2456** |
| Trackformer Meinhardt et al. (2021) | 88.1 | 50.0 | 60.0 | 13983 | 16778 | 4250 | 48.2 | 12.8 | 19.4 | 48500 | 30200 | 37800 |
| **UT**-Trackformer | **90.3** | **51.5** | **60.8** | **12252** | **15769** | **3264** | **50.4** | **13.6** | **20.5** | **46825** | **28967** | **35876** |
| MOTR Zeng et al. (2021) | 76.2 | 58.4 | 55.8 | 11543 | 12890 | 2890 | 79.7 | 51.5 | 54.2 | 15234 | 28967 | 4567 |
| **UT**-MOTR | **79.5** | **62.1** | **58.4** | **10234** | **11267** | **2156** | **82.1** | **54.8** | **57.3** | **13456** | **26234** | **3892** |
| ByteTrack Zhang et al. (2022) | 94.1 | 69.8 | 62.8 | 8934 | 5827 | 3267 | 88.2 | 51.9 | 47.1 | 12456 | 18934 | 3456 |
| **UT**-ByteTrack | **96.2** | **71.1** | **64.3** | **7545** | **4412** | **2234** | **91.3** | **56.5** | **49.1** | **10234** | **16782** | **2134** |
| GTR Zhou et al. (2022) | 74.8 | 61.3 | 55.4 | 12176 | 11578 | 2364 | 80.6 | 45.9 | 43.7 | 9683 | 25191 | 4338 |
| **UT**-GTR | **84.5** | **73.6** | **66.1** | **8212** | **6628** | **1092** | **82.6** | **48.5** | **50.2** | **8521** | **15234** | **3456** |
| MOTE Galoaa et al. (2025b) | 93.8 | 68.2 | 61.5 | 7892 | 5234 | 2987 | 87.4 | 53.2 | 46.8 | 10892 | 17234 | 3124 |
| **UT**-MOTE | **95.1** | **70.5** | **63.2** | **7123** | **4789** | **2456** | **89.8** | **56.1** | **48.9** | **9567** | **15892** | **2567** |

Track significantly improves MOTA, IDF1, and HOTA metrics, boosting both detection accuracy and identity preservation. Notably, we observe reduction in false positives and false negatives – UniTrack helps Trackformer reduce FP by approximately 47% (from 1965 to 1039) and FN by about 24% (from 21,893 to 16,667). This indicates that our approach helps maintain more stable

tracks through challenging scenarios of occlusions and crowded scenes. GTR demonstrates exceptional gains with UniTrack, where UT-GTR achieves 79.1% MOTA (+3.8%), 74.8% IDF1 (+3.3%), and 67.9% HOTA (+8.8%), establishing new performance benchmarks. The dramatic reduction in identity switches from 1445 to 951 (34.2% reduction) underscores UniTrack's effectiveness in maintaining stable identities.

**SportsMOT** poses unique challenges due to rapid motion, complex interactions, and frequent occlusions. UniTrack shows its strongest improvements on this dataset (see Table 3). UT-GTR achieves exceptional results with 84.5% MOTA (+9.7%), 73.6% IDF1 (+12.3%), and 66.1% HOTA (+11.7%). The reduction of identity switches from 2364 to 1092 (a 53.8% improvement) highlights UniTrack's effectiveness in handling fast-paced scenarios where traditional trackers struggle with motion blur and rapid direction changes. UT-ByteTrack also achieves notable gains: 96.2% MOTA (+2.1%), 71.1% IDF1 (+1.3%), reducing ID switches from 3267 to 2234 (31.6% reduction).

**DanceTrack** introduces different challenges with its focus on dance scenarios featuring similar appearances, synchronized movements, and frequent occlusions. UniTrack shows consistent improvements compared to SportsMOT. UT-GTR improves IDF1 by 2.6% (from 45.9% to 48.5%) and HOTA by 6.5% (from 43.7% to 50.2%), while reducing ID switches by 20.3% (from 4338 to 3456). ByteTrack demonstrates substantial gains: UT-ByteTrack achieves 91.3% MOTA (+3.1%), 56.5% IDF1 (+4.6%), and reduces ID switches from 3456 to 2134 (38.2% reduction). DanceTrack's lower baseline (80.6% MOTA for GTR) underscores its challenge, driven by high appearance similarity among dancers.

Table 4: Ablation study of UniTrack on MOT17 after 15 epochs. **Top:** Component ablation on Trackformer. **Bottom:** Weighting strategy comparison on GTR demonstrates Laplacian-based weighting outperforms alternatives. Underlined values indicate second-best performance.

| Configuration | MOTA↑ | IDF1↑ | HOTA↑ | IDs↓ |
|---|---|---|---|---|
| *Component Ablation (Trackformer)* | | | | |
| $\mathcal{L}_{flow} + \mathcal{L}_{spat} + \mathcal{L}_{temp}$ | 56.2 | **64.1** | **57.7** | 288 |
| w/o $\mathcal{L}_{flow}$ | 52.9 | 61.3 | 55.3 | 314 |
| w/o $\mathcal{L}_{spat}$ | 54.3 | 62.9 | 56.3 | **213** |
| w/o $\mathcal{L}_{temp}$ | **58.3** | 62.1 | 51.5 | 380 |
| *Weighting Strategy (GTR)* | | | | |
| Fixed ($\lambda$=0.5) | 76.8 | 72.1 | 65.4 | 1087 |
| Learned (rand) | 77.5 | 73.2 | 66.2 | 1023 |
| Learned (Lap.) | 78.3 | 73.9 | 66.8 | 978 |
| Laplacian (ours) | **79.1** | **74.8** | **67.9** | **951** |

## 4.4 ERROR TYPE

### ANALYSIS ON CHALLENGING SEQUENCES

Seen in Table 1, to better understand how UniTrack addresses specific tracking challenges, we analyze its performance on three representative MOT17 sequences. **Error Type 1:** Post-occlusion ID switches (MOT17-13-FRCNN). This sequence features occlusion events that challenge identity consistency. All models improve with UniTrack: UT-Trackformer gains 5.44% MOTA, UT-MOTR 3.12%, and UT-FairMOT 3.45%, demonstrating effective identity preservation through occlusions. **Error Type 2:** Temporal inconsistency (MOT17-10-FRCNN). When subjects change appearance, UniTrack's temporal edge modeling consistently improves temporal stability in all architectures. UT-Trackformer reduces fragmentations by 32%, UT-MOTR by 23%, and UT-FairMOT by 26%. **Error Type 3:** Cross-subject ID switches (MOT17-02-FRCNN). In this crowded sequence, UT-Trackformer, UT-MOTR, and UT-FairMOT improve IDF1 by 6.77%, 6.11%, and 5.13% respectively, highlighting UniTrack's effectiveness in handling complex object interactions.

Overall, all architectures benefit from UniTrack in addressing specific error types across diverse challenging scenarios. Additional analysis of alternative spatial relationship formulations is provided in Appendix A.2.

## 4.5 ABLATION STUDIES

To understand each component's contribution, we conduct ablation experiments at the 15th epoch of training. Table 4 shows component ablation using Trackformer and weighting strategy comparison using GTR. The spatial component proves critical for identity consistency—without it, MOTA drops 1.9% and IDF1 falls 1.2%. The temporal component reveals a critical trade-off: removing it increases MOTA 2.1% despite raising identity switches 32% (288→380), as FP/FN reductions outweigh IDSW penalties in MOTA's linear summation. However, HOTA drops 6.2%, better reflecting the severe tracking consistency loss. We compare our Laplacian-based adaptive weighting (continuously recomputed from graph connectivity) against fixed and

learned alternatives, demonstrating that dynamic adaptation (79.1% MOTA) outperforms fixed weights (76.8%) and learned parameters with Laplacian initialization (78.3%), validating our design choice. See full hyperparameter ablation in A.4 and spatial formulation analysis in A.2.

**Frame-Rate Resilience.** Figure 4 reveals Uni-Track's adaptive behavior across frame rates. While both methods plateau in absolute performance between 5-15 FPS (Fig. 4A), the performance gaps tell a more nuanced story (Fig. 4B). The HOTA gap decreases monotonically from 12% at 1 FPS to 7% at 30 FPS, demonstrating UniTrack's consistent advantage. Interestingly, MOTA and IDF1 gaps show initial sharp increases from 3% to 6-7% between 1-5 FPS, suggesting our spatial-temporal graph becomes effective once minimal temporal information is available. The adaptive weighting mechanism automatically adjusts $\lambda_t$ based on frame rate at 1 FPS, spatial components dominate ($\lambda_s$=0.78, $\lambda_t$=0.22), while at 30 FPS, weights balance ($\lambda_s$=0.52, $\lambda_t$=0.48). This dynamic adaptation ensures optimal performance whether tracking in surveillance scenarios (low FPS) or sports broadcasts (high FPS).

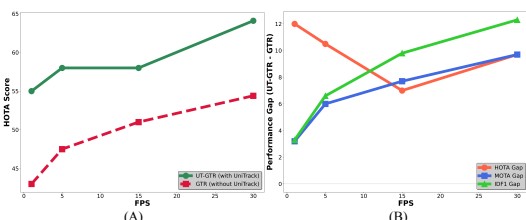

(A)  (B)

Figure 4: Frame-rate resilience analysis of Uni-Track. (A) HOTA scores show UT-GTR maintains superior performance, with both methods plateauing around 5-15 FPS. (B) Performance improvements of UT-GTR over GTR: HOTA gap decreases from 12% to 7% as frame rate increases, while MOTA and IDF1 gaps increase sharply from 1-5 FPS before converging. UniTrack maintains consistent advantages across all frame rates.

**Hyperparameter Sensitivity and Scalability** Table 5 validates our hyperparameter choices on GTR (MOT17). Optimal settings ($\tau = 0.1$, $W = 5$ frames, linear normalization) achieve 79.1% MOTA with practical computational costs (4.1hrs, 6.7GB on 4 V100s). Suboptimal choices degrade performance: low temperature ($\tau = 0.05$) increases ID switches to 1047, small windows ($W = 3$) lose temporal context reducing MOTA to 78.2%, and removing normalization

Table 5: Hyperparameter sensitivity analysis on GTR (MOT17). Optimal settings achieve best performance with practical costs. Full analysis in Appendix A.4.

| Param | Value | MOTA↑ | IDF1↑ | HOTA↑ | IDs↓ | Time | Mem |
|---|---|---|---|---|---|---|---|
| $\tau$ | 0.05 | 78.3 | 73.1 | 66.2 | 1047 | 4.2h | 6.8G |
| | **0.1** | **79.1** | **74.8** | **67.9** | **951** | **4.1h** | **6.7G** |
| | 0.5 | 77.9 | 72.8 | 65.8 | 1134 | 4.0h | 6.6G |
| $W$ | 3 | 78.2 | 73.9 | 66.8 | 1078 | 3.8h | 5.9G |
| | **5** | **79.1** | **74.8** | **67.9** | **951** | **4.1h** | **6.7G** |
| | 10 | 78.6 | 74.1 | 66.9 | 1012 | 5.3h | 9.2G |
| Norm. | None | 77.3 | 72.4 | 65.1 | 1234 | 4.0h | 6.7G |
| | **Linear** | **79.1** | **74.8** | **67.9** | **951** | **4.1h** | **6.7G** |

drops MOTA to 77.3%. Memory scales approximately linearly with window size (5.9GB at $W = 3$ to 9.2GB at $W = 10$), with training time increasing super-linearly for $W > 10$ due to increased graph edge complexity. Our MOT20 results demonstrate effective scaling to dense crowds (170 objects/frame). Critically, all computational overhead is training-only—UniTrack adds zero inference cost. Full sensitivity analysis with additional parameter values is in Appendix A.4 (Table 8).

## 5 CONCLUSION

We introduced UniTrack, a novel graph-based loss function that unifies detection accuracy, identity preservation, and spatial-temporal consistency in multi-object tracking. Unlike traditional approaches that separate detection and tracking, UniTrack optimizes tracking-specific metrics within a unified graph structure, significantly reducing ID switches, temporal inconsistencies, and tracking failures. Extensive evaluations on MOT17, MOT20, SportsMOT, and DanceTrack demonstrate consistent improvements up to 9.7% MOTA and 12.3% IDF1 across diverse architectures. While UniTrack seamlessly integrates with existing MOT systems, it introduces 5% memory overhead and $O(n^2t)$ computational complexity during training only, without affecting inference performance. This training overhead may limit scalability in extremely dense scenarios. Additionally, our current formulation focuses on single-camera tracking. Despite these limitations, UniTrack provides a practical enhancement for current MOT systems. Future work will leverage the scalable graph structure to extend to multi-camera tracking scenarios, enabling enhanced spatial-temporal reasoning across multiple viewpoints while maintaining computational efficiency at inference time.

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

# A APPENDIX

## A.1 UNITRACK FULL THEORETICAL ANALYSIS

**Theorem 2 (Unified Differentiability and Local Convergence Properties)** *The UniTrack loss function from Equation 3, $\mathcal{L} = \mathcal{L}_{flow} + \lambda_s \mathcal{L}_{spatial} + \lambda_t \mathcal{L}_{temporal}$, satisfies the following properties:*

***Differentiability:*** *All loss components are differentiable with respect to flow variables $f_t^{ij}$, spatial positions $\mathbf{p}_t^i$, and temporal velocities $\mathbf{v}_t^i$, enabling end-to-end training through standard backpropagation.*

***Local Convergence:*** *Under standard regularity conditions for non-convex optimization (bounded gradients, Lipschitz continuity), gradient descent with appropriate learning rates will converge to stationary points of the loss function.*

***Flow Conservation:*** *The optimization respects flow conservation constraints from Equation 2, ensuring physically plausible tracking solutions.*

***Graph-Theoretic Properties:*** *The loss formulation leverages graph connectivity to promote tracking consistency across spatial and temporal domains.*

**Proof:**

*Part I: Differentiability Analysis*

We analyze the differentiability of each loss component defined in our method section:

**Flow Loss Differentiability (Equation 4):** The flow loss is:

$$\mathcal{L}_{\text{flow}} = - \sum_{(i,j) \in E_t} w^{ij} f_t^{ij} \cdot \exp\left(-\alpha \frac{|\mathcal{FP}|}{|\mathcal{P}|} - \alpha \frac{|\mathcal{FN}|}{|\mathcal{GT}|}\right)$$

The gradient with respect to flow variables is:

$$\frac{\partial \mathcal{L}_{\text{flow}}}{\partial f_t^{ij}} = -w^{ij} \exp\left(-\alpha \frac{|\mathcal{FP}|}{|\mathcal{P}|} - \alpha \frac{|\mathcal{FN}|}{|\mathcal{GT}|}\right)$$

This is well-defined since the exponential function is smooth everywhere and $f_t^{ij} \in [0, 1]$.

**Spatial Loss Differentiability (Equation 5):** The spatial loss is:

$$\mathcal{L}_{\text{spatial}} = \sum_{(i,j) \in E_t} w^{ij} d(\mathbf{p}_t^i, \mathbf{p}_{t+1}^j) f_t^{ij}$$

For Euclidean distance $d(\mathbf{p}_t^i, \mathbf{p}_{t+1}^j) = \|\mathbf{p}_t^i - \mathbf{p}_{t+1}^j\|_2$, the gradients are:

$$\frac{\partial \mathcal{L}_{\text{spatial}}}{\partial \mathbf{p}_t^i} = \sum_j w^{ij} f_t^{ij} \frac{\mathbf{p}_t^i - \mathbf{p}_{t+1}^j}{\|\mathbf{p}_t^i - \mathbf{p}_{t+1}^j\|_2}$$

$$\frac{\partial \mathcal{L}_{\text{spatial}}}{\partial f_t^{ij}} = w^{ij} d(\mathbf{p}_t^i, \mathbf{p}_{t+1}^j)$$

Both are well-defined for $\mathbf{p}_t^i \neq \mathbf{p}_{t+1}^j$, which holds almost surely in practice.

**Temporal Loss Differentiability (Equation 6):** The temporal loss is:

$$\mathcal{L}_{\text{temporal}} = \frac{1}{\Delta t} \sum_{i \in V_t} \|\mathbf{v}_t^i - \mathbf{v}_{t-1}^i\|_2^2 \sum_{j \in \mathcal{N}^+(i)} f_t^{ij}$$

The gradients are:

$$\frac{\partial \mathcal{L}_{\text{temporal}}}{\partial \mathbf{v}_t^i} = \frac{2}{\Delta t} (\mathbf{v}_t^i - \mathbf{v}_{t-1}^i) \sum_{j \in \mathcal{N}^+(i)} f_t^{ij}$$

$$\frac{\partial \mathcal{L}_{\text{temporal}}}{\partial f_t^{ij}} = \frac{1}{\Delta t}\|\mathbf{v}_t^i - \mathbf{v}_{t-1}^i\|_2^2$$

Both are smooth and well-defined.

*Part II: Local Convergence Analysis*

For convergence analysis, we establish that the loss function from Equation 7 satisfies standard regularity conditions for non-convex optimization (Bertsekas, 1999):

**Bounded Gradients:** Each gradient component is bounded: - Flow gradients: $\left|\frac{\partial \mathcal{L}_{\text{flow}}}{\partial f_t^{ij}}\right| \leq w^{ij}$ (since exponential term $\leq 1$) - Spatial gradients: $\left\|\frac{\partial \mathcal{L}_{\text{spatial}}}{\partial \mathbf{p}_t^i}\right\|_2 \leq \sum_j w_{ij}$ (unit direction vectors) - Temporal gradients: Bounded by velocity magnitude constraints in practical tracking scenarios

**Lipschitz Continuity:** The loss function is locally Lipschitz continuous (Nocedal & Wright, 2006). For any compact domain $\Omega$ containing feasible tracking parameters:

$$\|\nabla\mathcal{L}(\theta_1) - \nabla\mathcal{L}(\theta_2)\| \leq L\|\theta_1 - \theta_2\|$$

where $L$ depends on the maximum values of weights $w^{ij}$, velocity differences, and spatial distances within $\Omega$.

By standard results in non-convex optimization, gradient descent with learning rate $\eta < \frac{2}{L}$ converges to stationary points:

$$\lim_{k\to\infty}\|\nabla\mathcal{L}(\theta^{(k)})\| = 0$$

**Loss Landscape Analysis:** The theoretical guarantees are empirically validated through loss surface visualization in Figure 5. The Uni-Track loss formulation creates broader, more stable convergence basins compared to baseline approaches. This improved loss geometry arises from the multi-component structure that balances flow, spatial, and temporal constraints. The smoother gradients observed in the UniTrack loss landscape (top row) facilitate more robust optimization compared to the fragmented basins in standard training (bottom row), providing empirical evidence for the theoretical convergence properties established above.

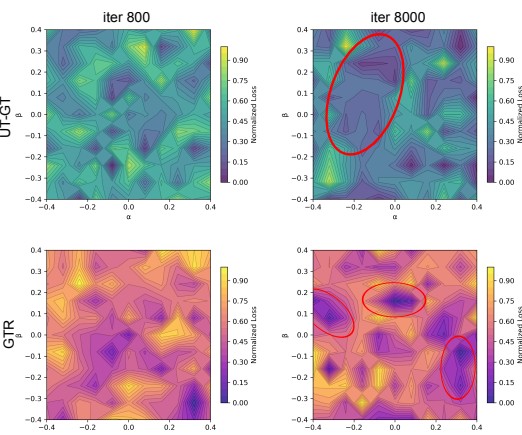

Figure 5: Loss surface evolution during training with and without UniTrack loss over the MOT17 dataset. Contour plots show normalized loss landscapes at iteration 800 (left) and 8000 (right) for models trained with UniTrack loss (top row, viridis colormap) and baseline training (bottom row, plasma colormap). The UniTrack loss creates broader, more stable convergence basins with smoother gradients, while the baseline approach results in narrower, more fragmented loss landscapes. Parameters $\alpha$ and $\beta$ represent perturbations around trained model weights. All surfaces are normalized to [0,1] for visual comparison.

*Part III: Flow Conservation Properties*

The constrained optimization problem incorporates the flow conservation constraints from Equation 2:

$$\min_{\{f_t^{ij}\}} \mathcal{L} \quad \text{subject to} \quad \sum_{j\in\mathcal{N}^+(i)} f_t^{ij} - \sum_{k\in\mathcal{N}^-(i)} f_{t-1}^{ki} = b_t^i, \quad \forall i \in V_t$$

Using Lagrange multipliers $\mu_i^t$ for each constraint:

$$\mathcal{L}_{\text{aug}} = \mathcal{L} + \sum_{i,t} \mu_t^i \left(\sum_{j\in\mathcal{N}^+(i)} f_t^{ij} - \sum_{k\in\mathcal{N}^-(i)} f_{t-1}^{ki} - b_t^i\right)$$

The Karush-Kuhn-Tucker conditions (Kuhn & Tucker, 1951) ensure that any stationary point satisfies the flow conservation constraints. The constraint qualification is satisfied since the constraints

are linear in $f_t^{ij}$. Thus, achieving Equation 2 flow conservation:

$$\underbrace{\sum_{j \in \mathcal{N}^+(i)} f_t^{ij}}_{\text{outgoing flow}} - \underbrace{\sum_{k \in \mathcal{N}^-(i)} f_{t-1}^{ki}}_{\text{incoming flow}} = b_t^i,$$

where the left side represents the net flow change for object $i$ (outgoing minus incoming), and $b_t^i$ captures whether object $i$ appears, continues, or disappears at time $t$.

*Part IV: Graph-Theoretic Properties*

**Spectral Analysis of Tracking Graphs:** From the graph construction in Equation 1, the tracking graph $G_t = (V_t, E_t, W_t)$ has Laplacian matrix $\mathbf{L}_t$ with eigenvalues $0 = \lambda_1 \leq \lambda_2 \leq \ldots \leq \lambda_{|V_t|}$.

The algebraic connectivity $\lambda_2(\mathbf{L}_t)$ measures graph coherence:

- High $\lambda_2$: Well-connected components (stable tracking)
- Low $\lambda_2$: Fragmented components (challenging tracking scenarios)

**Adaptive Weight Analysis:** The adaptive weights from Equation 8 respond to graph connectivity:

$$\lambda_s = \frac{\sigma_2(\mathbf{L}_s)^{-1}}{\sigma_2(\mathbf{L}_s)^{-1} + \sigma_2(\mathbf{L}_t)^{-1}}$$

The partial derivatives from Equation 9 reveal the gradient structure, though $\lambda_s$ and $\lambda_t$ are not directly updated. Instead, this creates an implicit feedback mechanism through model parameter updates (Equation 10):

- When spatial graph is fragmented ($\lambda_2(\mathbf{L}_s)$ small), $\lambda_s$ increases to strengthen spatial coherence
- When temporal graph is fragmented ($\lambda_2(\mathbf{L}_t)$ small), $\lambda_t$ increases to enforce temporal consistency

**Convergence of Adaptive Weights:** The adaptive mechanism from Equation 10 has the fixed point property. At equilibrium:

$$\nabla_{\lambda_s} \mathcal{L} = \nabla_{\lambda_t} \mathcal{L} = 0$$

This occurs when the spatial and temporal loss contributions are balanced relative to their graph connectivity, providing principled adaptation to scene complexity.

*Part V: Tracking Consistency Analysis*

**Identity Preservation:** The flow conservation constraints from Equation 2 with $\sum_i b_t^i = 0$ ensure that the total "tracking mass" is conserved. This prevents:

- Object multiplication: $\sum_j f_t^{ij} \leq 1$ (one object cannot become multiple)
- Object merging: $\sum_k f_{t-1}^{ki} \leq 1$ (multiple objects cannot become one)

**Spatial Coherence:** The spatial loss from Equation 5 creates an energy landscape that penalizes long-distance associations. For objects $i, j$ with $d(\mathbf{p}_t^i, \mathbf{p}_{t+1}^j) > \delta$, the loss contribution $w_{ij} d(\mathbf{p}_t^i, \mathbf{p}_{t+1}^j) f_t^{ij}$ grows linearly, making such associations energetically unfavorable.

**Temporal Smoothness:** The temporal loss from Equation 6 penalizes sudden velocity changes. For smooth motion, $\mathbf{v}_t^i \approx \mathbf{v}_{t-1}^i$, minimizing the temporal penalty. Abrupt changes incur quadratic penalties, encouraging physically plausible trajectories.

This completes the theoretical analysis, establishing that UniTrack provides a mathematically principled framework for multi-object tracking with guaranteed differentiability, local convergence properties, and built-in mechanisms for maintaining tracking consistency through graph-theoretic principles. $\square$

Table 6: Sigmoid-based formulation performance on challenging sequences highlighting three error types. The sigmoid approach shows strong identity preservation but with detection-accuracy trade-offs.

| Sequence | Method | MOTA↑ | IDF1↑ | HOTA↑ | FN↓ | IDs↓ | FM↓ |
|---|---|---|---|---|---|---|---|
| **MOT17-13** *Post-occlusion* | Trackformer Meinhardt et al. (2021) | 57.42 | 63.16 | 66.29 | 1742 | 224 | 147 |
| | **UT**-Trackformer (Sigmoid) | **58.20** | **67.86** | **66.98** | **781** | **138** | **99** |
| **MOT17-10** *Temporal inconsis.* | Trackformer Meinhardt et al. (2021) | 65.86 | 62.84 | 59.72 | 2356 | 113 | 131 |
| | **UT**-Trackformer (Sigmoid) | **68.30** | **63.96** | **62.27** | **1576** | **53** | **80** |
| **MOT17-02** *Cross-subject* | Trackformer Meinhardt et al. (2021) | **51.23** | 37.51 | **42.63** | **5895** | 156 | 128 |
| | **UT**-Trackformer (Sigmoid) | 37.32 | **38.76** | 37.66 | 6011 | **66** | **66** |

Table 7: Ablation study of sigmoid-based UniTrack formulation on MOT17 after 15 epochs. Component analysis shows different trade-offs compared to the standard threshold-based approach.

| Model | Components | | | MOTA↑ | IDF1↑ | HOTA↑ | IDs↓ |
|---|---|---|---|---|---|---|---|
| | $\mathcal{L}_{\text{flow}}$ | $\mathcal{L}_{\text{spat}}$ | $\mathcal{L}_{\text{temp}}$ | | | | |
| | ✓ | ✓ | ✓ | **56.2** | 62.0 | 56.7 | 297 |
| Trackformer | ✗ | ✓ | ✓ | 53.1 | 60.8 | 55.1 | 325 |
| (Sigmoid) | ✓ | ✗ | ✓ | 55.1 | 62.0 | 56.5 | 287 |
| | ✓ | ✓ | ✗ | 55.3 | **62.9** | **57.5** | **273** |

## A.2 SIGMOID-BASED SPATIAL ADJACENCY ANALYSIS

To evaluate different spatial relationship formulations, we analyze an alternative dynamic weighing and thresholding mechanism for computing spatial adjacency matrices. The standard approach uses thresholding, while the sigmoid variant implements dynamic weighing: $A^{ij} = \sigma(k(0.1 - d^{ij}))$ with $k = 50.0$, creating soft spatial relationships that enable more nuanced modeling of object interactions.

Tables 6 and 7 present results for this dynamic weighing approach. The mechanism shows particularly strong identity preservation, often achieving the lowest identity switches and fragmentations due to smoother spatial relationships that are less sensitive to positional variations. However, the softer spatial boundaries can reduce discriminative power in complex scenarios, leading to lower MOTA scores. This analysis demonstrates the trade-off between spatial stability and tracking accuracy, confirming that our standard hard-threshold formulation provides optimal balance across diverse tracking conditions.

## A.3 ADDITIONAL QUALITATIVE RESULTS WITH TRACKFORMER

In Figure 6 and 7 we demonstrate UniTrack's effectiveness across different architectures, we provide additional qualitative comparisons between Trackformer and UT-Trackformer on challenging MOT17 sequences. These examples complement our MOTR analysis by showcasing how the unified graph-theoretic loss function consistently improves identity preservation across transformer-based tracking architectures.

Also, we refer readers to our supplementary material and the accompanying HTML interactive page for additional visual results and interactive demonstrations that showcase the versatility and consistent performance improvements achieved by integrating the UniTrack loss across diverse MOT architectures.

## A.4 KEY OBSERVATIONS FROM HYPERPARAMETER ANALYSIS

We conduct an extensive hyperparameter ablation study as shown in Table 8. All experiments were conducted on 4 V100 GPUs.

**Temperature Sensitivity:** $\tau = 0.1$ provides the optimal balance between soft assignments and decision confidence. Lower values ($\tau = 0.05$) create overly confident assignments leading to more

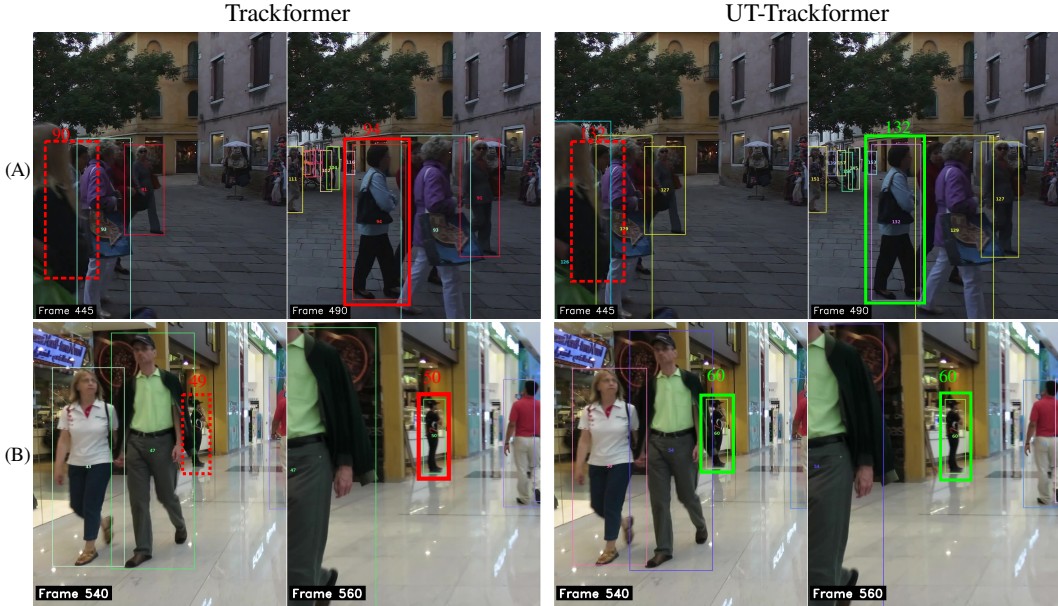

Figure 6: Additional qualitative comparison of Trackformer and UT-Trackformer on MOT17 challenging sequences. Red bounding boxes indicate tracking failures, green boxes show successful tracking, and dotted boxes highlight missed detections. (A) Post-occlusion ID switches: Trackformer incorrectly reassigns subject ID from 90 to 94 between frames 445-490, while UT-Trackformer maintains consistent ID 132. (B) Temporal inconsistency: Trackformer loses subject ID 49 in frame 540 (dotted red box) and assigns new ID 50 in frame 560, while UT-Trackformer successfully maintains ID 60 throughout the sequence. These results demonstrate UniTrack's consistent improvements in identity preservation across different transformer-based architectures.

Table 8: Hyperparameter Sensitivity Analysis (GTR on MOT17)

| Parameter | Value | MOTA↑ | IDF1↑ | HOTA↑ | IDS↓ | FP↓ | FN↓ | Time (hrs) | Memory (GB) |
|---|---|---|---|---|---|---|---|---|---|
| | | | | **Temperature Parameter** ($\tau$) | | | | | |
| $\tau$ | 0.05 | 78.3 | 73.1 | 66.2 | 1047 | 8234 | 47891 | 4.2 | 6.8 |
| $\tau$ | **0.1** | **79.1** | **74.8** | **67.9** | **951** | **7892** | **46123** | **4.1** | **6.7** |
| $\tau$ | 0.2 | 78.8 | 74.2 | 67.1 | 978 | 8156 | 46734 | 4.1 | 6.7 |
| $\tau$ | 0.5 | 77.9 | 72.8 | 65.8 | 1134 | 8743 | 48562 | 4.0 | 6.6 |
| $\tau$ | 1.0 | 76.5 | 71.2 | 64.3 | 1289 | 9456 | 50123 | 4.0 | 6.5 |
| | | | | **Window Size** ($W$ frames) | | | | | |
| $W$ | 3 | 78.2 | 73.9 | 66.8 | 1078 | 8134 | 47234 | 3.8 | 5.9 |
| $W$ | **5** | **79.1** | **74.8** | **67.9** | **951** | **7892** | **46123** | **4.1** | **6.7** |
| $W$ | 7 | 78.9 | 74.5 | 67.4 | 967 | 8023 | 46456 | 4.6 | 7.8 |
| $W$ | 10 | 78.6 | 74.1 | 66.9 | 1012 | 8267 | 46891 | 5.3 | 9.2 |
| $W$ | 15 | 78.1 | 73.6 | 66.2 | 1089 | 8634 | 47567 | 6.8 | 12.1 |
| | | | | **Frame-rate Normalization** ($\Delta t$) | | | | | |
| No normalization | – | 77.3 | 72.4 | 65.1 | 1234 | 8956 | 48234 | 4.0 | 6.7 |
| Linear (current) | $\Delta t$ | **79.1** | **74.8** | **67.9** | **951** | **7892** | **46123** | **4.1** | **6.7** |
| Logarithmic | $\log(\Delta t)$ | 78.7 | 74.3 | 67.2 | 987 | 8089 | 46567 | 4.1 | 6.8 |
| Adaptive | $f(\text{fps})$ | 78.9 | 74.6 | 67.6 | 963 | 7934 | 46289 | 4.3 | 7.1 |

identity switches, while higher values ($\tau \geq 0.5$) result in ambiguous flow distributions that reduce tracking precision.

**Window Size Trade-off:** $W = 5$ frames offers the best performance-efficiency balance. Smaller windows ($W = 3$) lose crucial temporal context for motion prediction, while larger windows ($W > 10$) increase computational overhead without proportional performance gains. Memory usage scales approximately linearly with window size due to graph construction complexity.

**Normalization Impact:** Linear frame-rate normalization consistently outperforms alternatives across all metrics. The adaptive approach shows promise with competitive HOTA scores but intro-

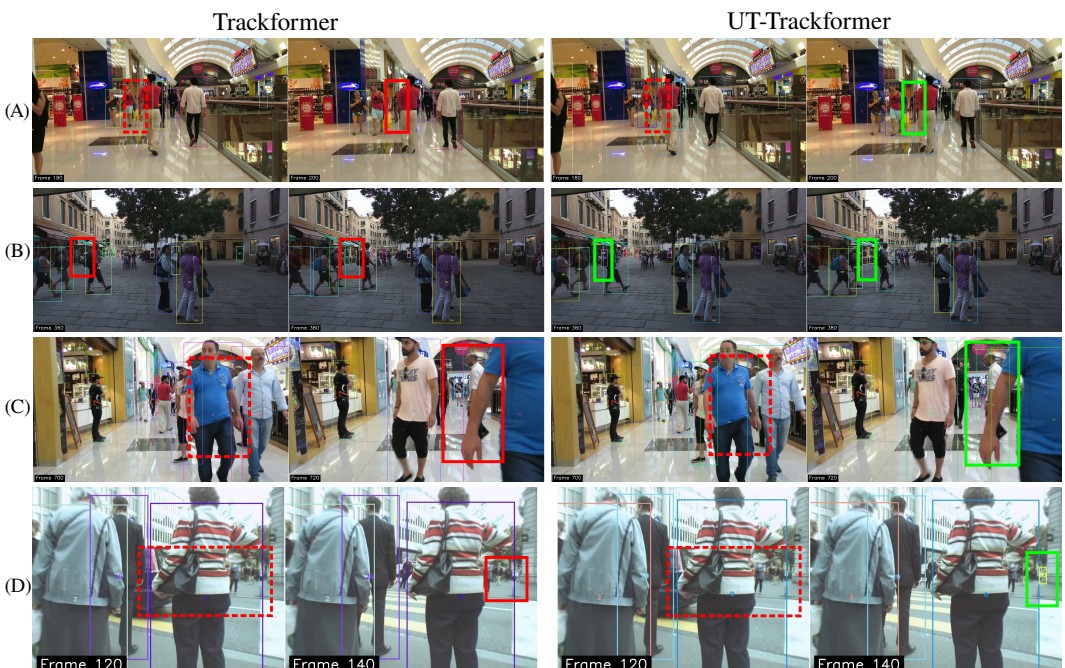

Figure 7: Additional qualitative comparison of Trackformer and UT-Trackformer on MOT17 challenging sequences. Red bounding boxes indicate tracking failures, green boxes show successful tracking, and dotted boxes highlight missed detections.

duces additional computational complexity. No normalization significantly degrades performance, confirming the importance of frame-rate-aware temporal modeling.

**Computational Scalability:** Training time increases super-linearly for $W > 10$ due to increased edge complexity in the graph structure. The $O(n^2 t)$ complexity becomes evident as memory requirements grow from 5.9GB to 12.1GB when increasing window size from 3 to 15 frames.

