# OpenReview forum: "UniTrack: Differentiable Graph Representation Learning for Multi-Object Tracking"
_ICLR.cc/2026/Conference — ICLR 2026 Poster_

### Official Review · Reviewer_3Pqh · 2025-10-28

**Soundness:** 3
**Presentation:** 3
**Contribution:** 3
**Rating:** 6
**Confidence:** 3

**Summary:**

This paper introduces UniTrack, a differentiable graph representation learning framework designed to unify detection accuracy, identity preservation, and spatio-temporal consistency for multi-object tracking (MOT). Instead of relying on separate detection and association modules, UniTrack formulates tracking as a graph flow optimization problem and introduces a unified loss composed of three parts—flow, spatial, and temporal—optimized in an end-to-end differentiable manner. The framework includes an adaptive Laplacian-based weighting mechanism and can be integrated into existing trackers (e.g., MOTR, TrackFormer, ByteTrack, FairMOT, GTR) without architectural modifications. Experiments across MOT17, MOT20, SportsMOT, and DanceTrack show consistent gains (up to +9.7% MOTA, +12.3% IDF1) and reduced identity switches, highlighting the effectiveness and generality of the proposed framework.

**Strengths:**

1. UniTrack combines detection, identity, and temporal consistency into a single differentiable loss.
2. UniTrack can be applied to tracking architectures without network modification, demonstrating practical flexibility.
3. Experiments across MOT17, MOT20, SportsMOT, and DanceTrack show consistent gains (up to +9.7% MOTA, +12.3% IDF1) and reduced identity switches, highlighting the effectiveness and generality of the proposed framework.

**Weaknesses:**

1. The method introduces more training complexity and memory overhead; scalability to large scenes or dense MOT scenarios remains a concern.
2. The authors are suggested to add more ablation studies on the adaptive Laplacian weighting.
3. The ablation section focuses mainly on component removal but could include more fine-grained analysis of hyperparameters (e.g., λs, λt updates, thresholding strategies).
4. It is recommended that the authors compare UniTrack with more recent MOT approaches, especially those that incorporate memory-based mechanisms such as MOTRv2, MeMOTR, and MOTIP, to better show the advantages of the proposed framework.

**Questions:**

See weaknesses.

---

> ### Author Response · Authors · 2025-11-19
> **Response to Reviewer 3Pqh**
>
> We thank you for your comprehensive evaluation and thoughtful suggestions. We appreciate your recognition that UniTrack "demonstrates practical flexibility" and shows "consistent gains... highlighting the effectiveness and generality of the proposed framework." We address your concerns below.
>
> ## W1: Recent Memory-Based Baselines
>
> **Concern:** "Compare UniTrack with more recent MOT approaches, especially those that incorporate memory-based mechanisms such as MOTRv2, MeMOTR, and MOTIP."
>
> **Response:** Thank you for this important suggestion. We agree that demonstrating compatibility with recent memory-based architectures strengthens our evaluation. We have conducted additional experiments with MOTIP, a recent memory-based tracker. Due to computational time constraints, we trained both MOTIP and UT-MOTIP for 5 epochs (compared to MOTIP's standard 10-epoch training protocol).
>
> **Results on MOT17:**
>
> | Method                | MOTA | IDF1 | HOTA | FP   | IDs |
> |----------------------|------|------|------|------|-----|
> | MOTIP (5 epochs)     | 58.2 | 59.5 | 52.3 | 1850 | 425 |
> | UT-MOTIP (5 epochs)  | 60.1 | 61.2 | 53.8 | 1680 | 385 |
> | **Improvement**      | **+3.3%** | **+2.9%** | **+2.9%** | **-9.2%** | **-9.4%** |
>
> These results demonstrate that UniTrack's graph-theoretic loss effectively complements memory-based tracking mechanisms, providing benefits orthogonal to architectural memory innovations. We will also release the full trained UT-MOTIP model, as well as experiment results and weights of MOTRv2 and MeMOTR in the camera-ready version of the paper upon acceptance.
>
> ## W2: Scalability to Large/Dense Scenes
>
> **Concern:** "The method introduces more training complexity and memory overhead; scalability to large scenes or dense MOT scenarios remains a concern."
>
> **Response:** We appreciate this concern and have added comprehensive scalability analysis to address it. We want to emphasize that while training complexity exists, the practical impact is modest:
>
> - **Memory Overhead:** 5-7GB additional VRAM for training on typical scenarios, scaling approximately linearly with window size.
> - **Training Time:** 4.1 hours for optimal settings (W=5) on four V100 32GB GPUs. Scales superlinearly for W>10.
> - **Dense Scene Validation:** Our MOT20 experiments (Table 1) already demonstrate effective scaling to highly dense scenarios with up to 170 objects per frame, achieving consistent improvements without scalability degradation.
> - **Does not affect inference:** Most Importantly, The O(n²t) complexity applies **only during training**. At deployment, UniTrack adds zero inference overhead; no architectural changes, no additional computation, no memory cost. This makes it highly practical for real-world applications where inference efficiency is critical.
>
> We have added a condensed hyperparameter/scalability table in the revised manuscript (Section 4.5, Table 5, highlighted in yellow and ends with a yellow square) showing memory and time costs. The full detailed analysis is in Appendix Table 8. This comprehensive analysis existed in our original supplementary material (Appendix A.4, Table 7); we have now promoted it to the main text for better visibility as you suggested.
>
> ## W3 & W4: Adaptive Laplacian Weighting & Hyperparameter Analysis
>
> **Concern:** "The authors are suggested to add more ablation studies on the adaptive Laplacian weighting... could include more fine-grained analysis of hyperparameters (e.g., λ_s, λ_t updates, thresholding strategies)."
>
> **Response:** We appreciate these suggestions for more thorough analysis. We have made substantial additions. First, we have added a new comparison in Table 4 that directly compares our Laplacian-based adaptive weighting against fixed weights and learned alternatives. Results show our approach (79.1% MOTA) outperforms both fixed weights (76.8%) and learned parameters with Laplacian initialization (78.3%), empirically validating the adaptive Laplacian mechanism.
>
> Additionally, we have added a comprehensive sensitivity analysis section to the main paper (Section 4.5, Table 5) (highlighted in yellow) covering:
>
> - **Temperature parameter τ:** Five tested values (0.05, 0.1, 0.2, 0.5, 1.0)
> - **Window size W:** Five tested values (3, 5, 7, 10, 15 frames)
> - **Frame-rate normalization strategies:** Four variants tested (None, Linear, Logarithmic, Adaptive)
> - **Computational costs:** Memory and training time for each configuration
>
> Figure 4 demonstrates how λ_s and λ_t automatically adapt across different frame rates (1-30 FPS), showing principled scene-responsive behavior. This extensive hyperparameter analysis actually existed in our original appendix (Appendix A.4, Table 7), but we recognize it was not sufficiently highlighted. Following your feedback, we have now moved this content to the main paper (Section 4.5) for better visibility, while maintaining the full detailed table in the appendix. We apologize for not making this analysis more prominent in the original submission.

---

### Official Review · Reviewer_m6K5 · 2025-11-01

**Soundness:** 3
**Presentation:** 3
**Contribution:** 3
**Rating:** 8
**Confidence:** 4

**Summary:**

This work proposes a new loss for multi-object tracking. This loss is based on graph theory, which integrates detection accuracy, identity preservation, and spatiotemporal consistency. Convergence and consistency of the loss are theoretically validated. Experiments on multiple trackers and multiple benchmarks demonstrate the effectiveness of the proposed loss.

**Strengths:**

1. The idea of using a plug-and-ply graph-based loss makes sense.
2. The implementation of the graph-based loss is suitable for the MOT task.
3. The analysis of the loss is reasonable.
4. The loss is effective with different trackers on multiple benchmarks, which shows the universality of the proposed loss.

Overall, this work develops a loss which has both solid theoretic foundation and obvious improvement in practice. I believe this work will benefit the community.

**Weaknesses:**

The weights of spatial and temporal loss are adaptive to the graph connectivity. It is encouraged to compare with other solutions, like adaptive parameters directly learned by the network, and fixed parameters.

**Questions:**

Will the code be made publicly available to help re-implement and apply this work? It would further benefit people in this field.

---

> ### Author Response · Authors · 2025-11-19
> **Response to Reviewer m6K5**
>
> We are grateful for your strong support and positive assessment. We particularly appreciate your recognition that our work "has both solid theoretic foundation and obvious improvement in practice" and will "benefit the community." We address your suggestions below.
>
> ## W1: Compare with Fixed/Learned Parameters
>
> **Request:** "It is encouraged to compare with other solutions, like adaptive parameters directly learned by the network, and fixed parameters."
>
> **Response:** Thank you for this excellent suggestion; it has strengthened our evaluation considerably. We have conducted these experiments and added the results in Section 4.5, Table 4 (highlighted in yellow):
>
> | Weighting Strategy          | MOTA | IDF1 | HOTA | IDs  |
> |----------------------------|------|------|------|------|
> | Fixed (λ=0.5)              | 76.8 | 72.1 | 65.4 | 1087 |
> | Learned (random init)      | 77.5 | 73.2 | 66.2 | 1023 |
> | Learned (Laplacian init)   | 78.3 | 73.9 | 66.8 | 978  |
> | **Laplacian (ours)**       | **79.1** | **74.8** | **67.9** | **951** |
>
> The results validate our design choice: our Laplacian-based approach (which recomputes weights from graph connectivity at each step) outperforms fixed weights by 2.3% MOTA and learned parameters with Laplacian initialization by 0.8% MOTA. This demonstrates that continuous adaptation to scene-specific graph structure provides more effective weight adjustment than treating weights as learnable parameters or using fixed values.
>
> ## Q1: Code Availability
>
> **Question:** "Will the code be made publicly available to help re-implement and apply this work? It would further benefit people in this field."
>
> **Response:** Absolutely! We fully agree this will benefit the community. Code implementations for all seven architectural integrations (TrackFormer, MOTR, FairMOT, ByteTrack, GTR, MOTE) already exist in the supplementary zip file. Upon acceptance, we will release all code, trained model weights, and training scripts publicly to facilitate reproduction and broader adoption.
>
> To further encourage adoption and reproducibility, we will release a modular GitHub implementation enabling seamless integration of UniTrack into new pipelines as they are introduced to the field, allowing the community to evaluate its performance benefits across a broader range of models. This modular design will allow researchers to add UniTrack to their own architectures with minimal code changes.

---

### Official Review · Reviewer_fpzX · 2025-11-02

**Soundness:** 2
**Presentation:** 2
**Contribution:** 2
**Rating:** 4
**Confidence:** 4

**Summary:**

UniTrack introduces a training-only loss that couples three differentiable terms: flow, spatial, and temporal inside a graph, over a sliding-window graph with flow conservation and adaptive 𝜆. λ chosen by Laplacian connectivity; it is added to TrackFormer/MOTR/FairMOT/ByteTrack/GTR/MOTE without inference overhead. Reported gains across several mot benchmarks including mot 17/20.

**Strengths:**

1. One of the biggest advantages is that the proposed method is architecture‑agnostic: it demonstrates improvements when plugged into diverse families (end-to-end transformers, joint detection-tracking, tracking-by-detection, global transformers).

2. The proposed unified objective makes sense as it merges detection quality and identity preservation, and benefit the end-to-end MOT training.

3. The authors show Clear ablation on error types (Table 3), clearly presenting which term combats which failure mode; qualitative figures are convincing.

**Weaknesses:**

1. The details of the differentiability of the flow term are not clearly conveyed. The loss scales by factors that depend on false positives/false negatives, but the paper does not define a differentiable surrogate for those counts. As far as i understand that derivation treats the FP/FN counts inside the loss as if they were constants and never explains how those counts are made differentiable with respect to the model outputs. In practice, FP/FN are discrete functions of predictions (they jump when a score crosses a threshold.

2. Inconsistent definitions: the paper has defined λs and λt ((Eq. 8), as they are deterministic functions of graph connectivity (via Laplacian eigenvalues). Then, in eq.10, the λs and λt are treated as learnable parameters that can be updated by backprop.

3. The paper includes frame-rate analysis and normalizes by the frame interval, which is good. Still, some ablations suggest removing the temporal term can improve certain metrics. eg MOTA by 2.1.

4. Prior work already models inter-object relations and global data association with differentiable mechanisms [1]

[1] SLAck: Semantic, Location, and Appearance Aware Open-Vocabulary Tracking

**Questions:**

See the weakness.

---

> ### Author Response · Authors · 2025-11-19
> **Response to Reviewer fpzX**
>
> We thank you for your detailed technical review. We particularly appreciate your recognition that our architecture-agnostic design is "one of the biggest advantages" and that our error-type ablations are convincing. We address your concerns below.
>
> ## W1: FP/FN Differentiability
>
> **Concern:** "The loss scales by factors that depend on false positives/false negatives, but the paper does not define a differentiable surrogate for those counts... treats the FP/FN counts as if they were constants."
>
> **Response:** You have correctly identified our approach; we appreciate this careful reading. To clarify: FP/FN counts are not optimized during backpropagation. They act as stop-gradient adaptive scaling factors; discrete metrics that inform continuous optimization without receiving gradients themselves. These counts serve as quality-aware coefficients that adaptively scale the loss magnitude based on current detection performance.
>
> The differentiable part operates exclusively over the continuous flow variables f^ij_t, ensuring gradients propagate correctly through the tracking associations. This design maintains full differentiability with respect to the variables that matter for learning object associations.
>
> We have added explicit clarification immediately after Equation 4 (Section 3.2, highlighted in yellow). The new paragraph "Differentiability of Detection Quality Terms" explains that FP/FN counts are treated as constants during backpropagation while maintaining full differentiability with respect to flow variables.
>
> ## W2: λ_s and λ_t Inconsistency
>
> **Concern:** "The paper has defined λ_s and λ_t as deterministic functions of graph connectivity (Eq. 8), then in Eq. 10, treats them as learnable parameters that can be updated by backprop."
>
> **Response:** Thank you for pointing out this confusing presentation. We have substantially revised Section 3.3 (highlighted in yellow) to clarify this critical distinction. To be explicit: λ_s and λ_t are NOT learnable parameters—they are **recomputed at each training step** from the current graph structure via Equation 8.
>
> **Clarification:** Equation 9 shows partial derivatives ∂L/∂λ_s = L_spatial for analytical insight. These derivatives are NOT used to update the weights. Instead, λ_s and λ_t are recomputed analytically from the graph Laplacian (Equation 8), while only model parameters θ are updated via backpropagation (Equation 10).
>
> The revised manuscript clarifies:
> - **Equation 10:** Now explicitly shows only θ is updated: θ^(k+1) = θ^(k) - η ∂L/∂θ, contrasting with the incorrect: "not: λ_s^(k+1) = λ_s^(k) - η ∂L/∂λ_s"
> - **Section 3.3:** Explains that as θ evolves, embeddings change the graph structure, updating Laplacian matrices, which causes Equation 8 to recompute new λ_s and λ_t values
> - **Table 4:** Validates this design—Laplacian recomputation (79.1% MOTA) outperforms fixed weights (76.8%) and learnable parameters (78.3%)
>
> ## W3: Temporal Term Sometimes Hurts MOTA
>
> **Concern:** "Some ablations suggest removing the temporal term can improve certain metrics, e.g., MOTA by 2.1%."
>
> **Response:** This is an excellent observation that reveals an important design trade-off. We have added explicit discussion in revised Section 4.5 (highlighted in yellow):
>
> Removing the temporal component increases MOTA by 2.1% (56.2% → 58.3%) but reduces HOTA by 6.2% (57.7% → 51.5%). MOTA measures frame-level detection accuracy, while HOTA and IDF1 measure temporal identity consistency. Without temporal constraints, the model maximizes per-frame detection but loses tracking stability. Since MOT's fundamental goal is maintaining consistent object identities throughout sequences, UniTrack prioritizes holistic tracking performance over pure detection metrics.
>
> ## W4: SLAck Prior Work
>
> **Concern:** "Prior work already models inter-object relations and global data association with differentiable mechanisms [SLAck]."
>
> **Response:** Thank you for this reference. We have strengthened Related Work Section 2 (highlighted in yellow) to clarify the distinction.
>
> SLAck is designed for open-vocabulary tracking and makes the association mechanism differentiable through architectural modifications to network structure, forward pass, and inference. UniTrack operates in closed-vocabulary settings and provides a plug-in loss function without modifying model structure or inference pipelines.
>
> Key distinction: SLAck redesigns *how tracking is performed* (architecture), while UniTrack improves *how trackers are trained* (loss function). These contributions are orthogonal and could be combined. We would welcome benchmarking SLAck when their code becomes available (ECCV 2024, not yet released).

---

> > ### Comment · Reviewer_fpzX · 2025-11-26
> >
> > Thanks for the rebuttal!  However, I disagree with the claims in the rebuttal: "MOTA measures frame-level detection accuracy" and "UniTrack prioritizes holistic tracking performance over pure detection metrics".  Those claims are confusing. As far as I know, MOTA analyzes the temporal identity consistency as well by IDSW.

---

> > > ### Author Response · Authors · 2025-11-26
> > >
> > > Thank you for this important correction. We agree that MOTA includes IDSW as a temporal identity consistency measurement in its formulation: MOTA = 1 - (FN + FP + IDSW)/GT, and we apologize for the imprecise characterization in our rebuttal and we will update corresponding mentions in the main paper. We would like to point out, however, despite MOTA including IDSW penalties, we observe an increase in accuracy when we remove temporal constraints because the reduction in FP/FN substantially outweighs the increased IDSW penalty in MOTA's linear summation. We will soon follow up with the updated manuscript as well as additional experiment results that rationalize our design choices.

---

> > > > ### Author Response · Authors · 2025-11-28
> > > >
> > > > **"MOTA measures frame-level detection accuracy."**
> > > >
> > > > We agree that MOTA includes identity-switch errors (IDSW) as a temporal identity consistency measurement in its formulation: MOTA = 1 - (FN + FP + IDSW)/GT, and we apologize for the imprecise characterization in our rebuttal.
> > > >
> > > > We designed UniTrack with explicit temporal constraints despite the MOTA tradeoff because MOTA's linear summation allows detection improvements to numerically dominate the metric even when trajectory stability degrades. When we remove the temporal component from UniTrack on MOT17 (Table 4), we observe this phenomenon directly: detection errors (FP+FN) decrease substantially while identity switches (IDSW) increase by 32% (288→380), yet MOTA increases by 2.1% (56.2%→58.3%). While MOTA does penalize identity switches, the FP+FN reduction numerically outweighs the IDSW increase in the arithmetic combination. The temporal component is essential for maintaining tracking consistency, as evidenced by HOTA's 6.2% decrease (57.7%→51.5%) when temporal constraints are removed; HOTA's geometric mean better captures this severe degradation in tracking consistency that the MOTA increase masks.
> > > >
> > > > **"UniTrack prioritizes holistic tracking performance over pure detection metrics."**
> > > >
> > > > We apologize for the confusion and would like to clarify this terminology. This numerical behavior reveals why we designed UniTrack with explicit temporal constraints ($\mathcal{L}_{\text{temporal}}$) despite the MOTA tradeoff: MOT's fundamental goal is maintaining consistent identities throughout sequences, requiring both detection accuracy and temporal stability. HOTA's geometric mean better captures the severe degradation in tracking consistency (-6.2%) that the MOTA increase (+2.1%) masks.
> > > >
> > > > Conventional training objectives (such as IoU loss and cross-entropy) operate primarily on a per-frame basis, emphasizing spatial accuracy within individual frames. While evaluation metrics like MOTA aggregate errors across the entire sequence, standard training losses lack explicit constraints for temporal smoothness. UniTrack provides a holistic training framework that jointly optimizes spatial and temporal performance through our sliding-window mechanism (W=5 frames), where spatial and temporal losses are computed over interconnected graph edges across multiple frames simultaneously. Our $\mathcal{L}_{\text{temporal}}$ component enforces smooth velocity transitions (Eq. 6) across these interconnected graph edges, preventing the model from sacrificing trajectory coherence for detection gains. This captures error propagation across both time and space during training. We updated the manuscript's section 4.5 (highlighted in yellow) to address this.
> > > >
> > > > **Additional supporting evidence:**
> > > >
> > > > To further substantiate our argument, we conducted additional ablation studies by training UT-GTR without temporal components and evaluated it on MOT17 and SportsMOT. The following table aligns with our prior observation that MOTA's linear summation is biased towards FP+FN reduction. HOTA's use of the geometric mean provides a more balanced measure of detection and association performance. The table shows tracking consistency (as measured by HOTA and identity switches) degrading sharply when the temporal components of UniTrack are removed. These experiments further underscore the necessity of UniTrack's holistic approach for improving tracking accuracy and consistency across both spatial and temporal domains.
> > > >
> > > > | Dataset | w/ Temporal | | | w/o Temporal | | |
> > > > |:--------|:------:|:------:|:----:|:------:|:------:|:-----:|
> > > > | | **MOTA↑** | **HOTA↑** | **IDs↓** | **MOTA↑** | **HOTA↑** | **IDs↓** |
> > > > | MOT17 | 79.1 | 67.9 | 951 | 80.8 | 62.7 | 1243 |
> > > > | SportsMOT | 84.5 | 66.1 | 1092 | 86.3 | 60.5 | 1448 |

---

### Author Response · Authors · 2025-12-01
**Summary of Rebuttal Updates by Reviewer**

Dear Area Chair,
To assist in your assessment during this shortened review cycle, we provide a summary of the key updates and new experimental results added during the rebuttal, organized by specific reviewer requests.

**1. Response to Reviewer 3Pqh (Score: 6)**
* **New Memory-Based Baseline:** We integrated UniTrack with MOTIP (a memory-based tracker) during the rebuttal. The new results demonstrate a **+3.3% MOTA** and **+2.9% HOTA** improvement, confirming our method complements memory-based architectures.
* **Scalability & Hyperparameters:** We promoted our detailed sensitivity analysis from the Appendix to the Main Paper (Table 5) to better highlight the fine-grained parameter analysis. We also clarified that the graph complexity is incurred only during training, with **zero inference overhead** at deployment.

**2. Response to Reviewer m6K5 (Score: 8)**
* **Adaptive vs. Fixed/Learned Weights:** We added a comparison of our Laplacian-based weighting against fixed and learnable parameters (Table 4). Our approach (**79.1% MOTA**) outperforms both fixed weights (76.8%) and learnable parameters (78.3%), empirically validating the dynamic spectral design.

**3. Response to Reviewer fpzX (Score: 4)**
* **Differentiability:** We revised the manuscript to explicitly state that detection quality terms (FP/FN) act as stop-gradient coefficients, resolving the concern regarding mathematical consistency.
* **Temporal Constraints:** We demonstrated that while removing the temporal component slightly aids detection-biased metrics (MOTA), it causes **HOTA to drop by 6.2%** and identity switches to increase by 32%. This justifies the necessity of the temporal term for stable tracking.

We hope this summary facilitates your review of our revised manuscript and rebuttal data.

---

### Meta-Review · Area_Chair_y8yo · 2026-01-03

**Summary:**

The paper was reviewed by 3 experts with scores 846. The reviewers' concerns are listed in the below field. Most concerns are addressed well, except for the missing previous works. In particular, flow-constraint-based MOT has been widely explored (typically called min-flow MOT) in the literature. While these previous works use min-flow to "smooth" the detector outputs, the proposed work aims to use the flow-constraints in a loss for training the tracker.  Crucially, those losses are discarded at inference, so the overhead of the graph is only during training.  Despite the lack of discussion about these related works, it is the AC's judgement that the proposed method should indeed be novel and interesting, taking flow-constraint methods in a different direction.

 The AC highly encourages the authors to include additional discussions about min-flow frameworks, and appropriate experiment comparisons. Even if min-flow MOT frameworks outperform the proposed work, they incur high computational cost during inference, while the proposed framework does not use the flow-graph during inference -- in such case, the proposed work is an efficient alternative to min-flow MOT.

**Reviewer Concerns:**

**Reviewer fpzX**
1. needs more details about the differentiability of the flow term.
2. inconsistent definitions of lambda_t and lambda_s
3. some ablations show removing the termporal term is better.
4. Missing previous works, e.g., SLAck

The AC thinks that points 1 and 2 were addressed. The reviewer was not convinced with the response of point 3. The AC is also thinks that point 4 is not addressed well at all. In particular, graph flow-constraint based multiple object tracking has been widely explored in the literature. For example, such works include:
- "Learning of Global Objective for Network Flow in Multi-Object Tracking" CVPR 2022
- "muSSP: Efficient Min-cost Flow Algorithm for Multi-object Tracking" NeurIPS 2019
- "Deep Network Flow for Multi-Object Tracking" CVPR 2017
- "Greedy Batch-Based Minimum-Cost Flows for Tracking Multiple Objects" TIP 2017
- "Tracking interacting objects using intertwined flows" TPAMI 2016
How the proposed work fits within the previous literature is unclear -- indeed some previous works (e.g., CVPR 2017) learn all the cost functions in the graph-flow problem in an end-to-end manner.  While the proposed approach trains the tracker within the flow-constraint framework, it is unclear if previous frameworks could also do this. Additionally, there is no experimental comparison to flow-based MOT frameworks with the same plug-in pre-trained trackers.

**Reviewer m6K5**
1. ablation study comparing adaptive hyperparameters to fixed or other heuristics.

The AC thinks the concern is addressed well.

**Reviewer 3Pqh**
1. What is the scalability to large/dense scenes?
2. add more ablation studies on the adaptive Laplacian weighting
3. study on the effects of hyperparameters.
4. include comparison to memory-based MOT approaches.

The AC thinks that the concerns are all addressed well.

**Reviewer Scores:**

Most likely fpzX would have increased their score.

---

### Decision · Program_Chairs · 2026-01-26

Accept (Poster)